# An Ecological Analysis of Hospitalization Patterns for Diseases of the Nervous System in England and Wales over the Last 20 Years

**DOI:** 10.3390/healthcare10091670

**Published:** 2022-09-01

**Authors:** Abdallah Y. Naser, Eman Zmaily Dahmash, Tamara Al-Daghastani, Hassan Alwafi, Sawsan Abu Hamdah, Zahra K. Alsairafi, Fatemah M. Alsaleh

**Affiliations:** 1Department of Applied Pharmaceutical Sciences and Clinical Pharmacy, Faculty of Pharmacy, Isra University, Amman 11622, Jordan; 2Department of Medical Allied Sciences, Al-Balqa Applied University, Al-Salt 19117, Jordan; 3Faculty of Medicine, Umm Al-Qura University, Mecca 21955, Saudi Arabia; 4College of Pharmacy, Department of Pharmaceutical Sciences, Al-Ain University, Abu Dhabi Campus, Abu Dhabi 112612, United Arab Emirates; 5Department of Biopharmaceutics and Clinical Pharmacy, Faculty of Pharmacy, The University of Jordan, Amman 11942, Jordan; 6Department of Pharmacy Practice, Faculty of Pharmacy, Kuwait University, Hawalli 25210, Kuwait

**Keywords:** admission, central nervous system, England, hospitalization, Wales

## Abstract

Objectives: This study aims to provide a comprehensive overview of the hospitalization pattern of nervous system diseases from 1999 to 2019. Methods: This is ecological research based on data from the Hospital Episode Statistics database in England and the Patient Episode Database in Wales, both of which are publicly available. Data on hospital admissions were collected between April 1999 and March 2019. Diagnostic codes (G00–G09: inflammatory diseases of the central nervous system, G10–G14: systemic atrophies primarily affecting the central nervous system, G20–G26: extrapyramidal and movement disorders, G30–G32: other degenerative diseases of the nervous system, G35–G37: demyelinating diseases of the central nervous system, G40–G47: episodic and paroxysmal disorders, G50–G59: nerve, nerve root and plexus disorders, G60–G65: polyneuropathies and other disorders of the peripheral nervous system, G70–G73: diseases of myoneural junction and muscle, G80–G83: cerebral palsy and other paralytic syndromes, and G89–G99: other disorders of the nervous system) from the tenth edition of the International Statistical Classification of Diseases and Related Health Problems 10th Revision (ICD-10) were used to identify hospital admissions. A Poisson model was used to examine the trend in hospital admissions. Results: During the study period, hospital admission rate increased by 73.5% (from 474.44 (95% CI 472.58–476.31) in 1999 to 823.37 (95% CI 821.07–825.66) in 2019 per 100,000 persons, trend test, *p <* 0.01). The most prevalent diseases of the nervous system hospital admissions causes were episodic and paroxysmal disorders, nerve, nerve root, and plexus disorders, and demyelinating diseases of the central nervous system which accounted for 37.4%, 22.1%, and 9.3%, respectively. Hospital admission rate between females increased by 79.1% (from 495.92 (95% CI 493.25–498.58) in 1999 to 888.33 (95% CI 884.97–891.68) in 2019 per 100,000 persons). Hospital admission rate between males was increased by 67.5% (from 451.88 (95% CI 449.28–454.49) in 1999 to 756.82 (95% CI 753.69–759.96) in 2019 per 100,000 persons). Conclusion: In the United Kingdom, hospital admissions for diseases of the nervous system are on the rise. Future research is needed to identify high-risk groups and suggest effective interventions to reduce the prevalence of these disorders.

## 1. Introduction

Globally, nervous system neurological disorders are leading cause of disability contributing to an overall health burden that is escalating [1]. In 2016, nervous system disorders were responsible for 276 million disability-adjusted life-years (DALYs), which comprised 11.6% of total DALYs from all diseases [2].

The main disorders encountered in the nervous system are listed by the International Statistical Classification of Diseases and Related Health Problems 10th Revision (ICD-10) are related to neurological disorders such as parkinsonism, dementia, epilepsy, Alzheimer’s, migraine, headache, stroke, nerve, nerve root, and plexus disorders, and demyelinating diseases of the central nervous system [3]. The WHO reported that the top five causes of years of healthy life lost due to disability (YLD) among neurological disorders in high-income countries were Alzheimer’s and other dementia, followed by cerebrovascular disease, neurological injuries, migraine, and neuropathies [4]. However, the rise in life expectancy and reduction in fertility resulted in a demographical shift from mainly youthful population into aging ones, triggering the increases in neurological disorders such as Alzheimer’s, dementia, and Parkinson’s disease [5,6,7].

Previous research in the United Kingdom looked at the patterns of hospitalization for various chronic and acute diseases. To our knowledge, no studies have looked at the pattern of hospital admissions for all nervous system diseases across all age groups over the last 20 years. In 2010, there were approximately 135 million individuals worldwide suffering from dementia [6]. This number resulted in a huge economic impact which with an annual estimate of 600 billion USD [7]. Since the prevalence of dementia progresses sharply at ages above 75, the projected increase in dementia cases by 2050 is three times the reported one [6,7]. As disorders of the nervous system, commonly reported as neurological disorders, are progressively identified as major causes of death and disability globally, this study aimed at providing a comprehensive analysis of the hospitalization pattern of such diseases during the period of 1999–2019 using publicly available data of England and Wales. 

## 2. Methods

### 2.1. Study Sources and the Population

This was an ecological study using publicly available data extracted from the Hospital Episode Statistics (HES) database in England [8] and the Patient Episode Database for Wales (PEDW) for the period between April 1999 and April 2019 [9]. The HES and PEDW databases contain hospital admission data for individuals with nervous system diseases of all ages, separated into four age groups: under 15 years, 15–59 years, 60–74 years, and 75 years and up. Using the ICD-10, we identified hospital admissions connected to nervous system diseases. All diagnosis codes for nervous system diseases ((G00–G09: inflammatory diseases of the central nervous system, G10–G14: systemic atrophies primarily affecting the central nervous system, G20–G26: extrapyramidal and movement disorders, G30–G32: other degenerative diseases of the nervous system, G35–G37: demyelinating diseases of the central nervous system, G40–G47: episodic and paroxysmal disorders, G50–G59: nerve, nerve root and plexus disorders, G60–G65: polyneuropathies and other disorders of the peripheral nervous system, G70–G73: diseases of myoneural junction and muscle, G80–G83: cerebral palsy and other paralytic syndromes, and G89–G99: other disorders of the nervous system) were used to identify all hospital admissions in England and Wales connected to various forms of nervous system diseases [3]. All hospital admissions, outpatients, and Accident and Emergency (A&E) activities undertaken at all National Health Service (NHS) trusts and any independent sector financed by NHS trusts are recorded in the HES and PEDW databases. From 1999/2000 onward, data on hospital admissions in England and Wales are available. Patient demographics, clinical diagnosis, procedures, and length of stay are among the information available. We used mid-year population data from the Office for National Statistics (ONS) to compute the annual hospital admission rate for nervous system diseases between 1999 and 2019 [10]. 

### 2.2. Statistical Analysis

Using the finished consultant episodes of diseases of the nervous system-related admission divided by the mid-year population, hospital admission rates with 95% confidence intervals (CIs) were determined. The Pearson Chi-square test for independence was used to estimate the variation in hospital admission rates between 1999 and 2019 because we were using independent frequency data with a sufficient sample size. 

A Poisson regression model with robust variance estimation was used to examine the trend in hospital admissions. We conducted three independent Poisson regression models with the admission rate as the dependent variable and the years of admission for the general population, gender (across the years), and age group (across the years) as the independent variables. SPSS version 27 was used for all analyses (IBM Corp., Armonk, NY, USA).

## 3. Results

The total number for diseases of the nervous system hospital admissions for different causes increased by 97.8% from 247,376 in 1999 to 489,408 in 2019, expressing an increase in hospital admission rate of 73.5% (from 474.44 (95% CI 472.58–476.31) in 1999 to 823.37 (95% CI 821.07–825.66) in 2019 per 100,000 persons, trend test, *p* < 0.01). 

The most prevalent diseases of the nervous system hospital admissions causes were episodic and paroxysmal disorders, nerve, nerve root, and plexus disorders, and demyelinating diseases of the central nervous system which accounted for 37.4%, 22.1%, and 9.3%, respectively (Table 1).

Throughout the past two decades, the hospital admission rate for diseases of the nervous system increased, except for cerebral palsy and other paralytic syndromes, which decreased. The most distinguished increase in the hospital admissions rate was seen in polyneuropathies and other disorders of the peripheral nervous system, demyelinating diseases of the central nervous system, inflammatory diseases of the central nervous system, other disorders of the nervous system, and diseases of myoneural junction and muscle 3.36-fold, 1.60-fold, 1.54-fold, 1.43-fold, and 1.36-fold, respectively (Table 2, Figure 1).

Concerning age group difference for diseases of the nervous system hospital admission, the age group 15–59 years accounted for 50.5% of the total number of diseases of the nervous system hospital admissions, followed by the age group 60–74 years with 21.1%, the age group 75 years and above with 19.8%, and then the age group below 15 years with 8.6%. Rates of hospital admission for diseases of the nervous system among patients aged below 15 years increased by 77.9% (from 228.54 (95% CI 225.57–231.51) in 1999 to 406.64 (95% CI 402.83–410.45) in 2019 per 100,000 persons). Rates of hospital admission for diseases of the nervous system among patients aged 15–59 years increased by 93.0% (from 378.38 (95% CI 376.23–380.53) in 1999 to 730.42 (95% CI 727.58–733.27) in 2019 per 100,000 persons). Rates of hospital admission for diseases of the nervous system among patients aged 60–74 years increased by 46.5% (from 752.78 (95% CI 746.34–759.21) in 1999 to 1102.88 (95% CI 1096.14–1109.61) in 2019 per 100,000 persons). Rates of hospital admission for diseases of the nervous system among patients aged 75 years and above increased by 29.6% (from 1371.13 (95% CI 1359.61–1382.65) in 1999 to 1777.42 (95% CI 1765.93–1788.91) in 2019 per 100,000 persons) (Figure 2).

A total of 7,380,741 for diseases of the nervous system hospital admission episodes were recorded in England and Wales during the study time. Females contributed to 54.0% of the total number of diseases of the nervous system hospital admission accounting for 3,984,642 hospital admission episodes by a mean of 199,232 per year. Hospital admission rate between females increased by 79.1% (from 495.92 (95% CI 493.25–498.58) in 1999 to 888.33 (95% CI 884.97–891.68) in 2019 per 100,000 persons). Hospital admission rate between males increased by 67.5% (from 451.88 (95% CI 449.28–454.49) in 1999 to 756.82 (95% CI 753.69–759.96) in 2019 per 100,000 persons) (Figure 3).

### 3.1. Admission Rates Stratified by Gender

Diseases of the nervous system hospital admission rates for inflammatory diseases of the central nervous system, systemic atrophies primarily affecting the central nervous system, extrapyramidal and movement disorders, episodic and paroxysmal disorders, polyneuropathies and other disorders of the peripheral nervous system, diseases of myoneural junction and muscle, and cerebral palsy and other paralytic syndromes were higher among males compared to females, while diseases of the nervous system hospital admission rates for other degenerative diseases of the nervous system, demyelinating diseases of the central nervous system, nerve, nerve root, and plexus disorders, and other disorders of the nervous system were higher among females compared to males (*p* < 0.05) (Figure 4).

### 3.2. Admission Rates Stratified by Age Group

The rates of diseases of the nervous system-related hospital admissions were seen to be directly related to age. That includes the following: systemic atrophies primarily affecting the central nervous system, extrapyramidal and movement disorders, other degenerative diseases of the nervous system, episodic and paroxysmal disorders, nerve, nerve root, and plexus disorders, and other disorders of the nervous system. Additionally, hospital admissions due to diseases of myoneural junction and muscle were more prevalent among the age group: 75 years and above, 60–74 years, below 15 years, and 15–59 years, in that order. Hospital admissions due to cerebral palsy and other paralytic syndromes were more prevalent among the age group: 75 years and above, below 15 years, 60–74 years, and 15–59 years, in that order. Hospital admissions due to polyneuropathies and other disorders of the peripheral nervous system were more prevalent among the age group: 60–74 years, 75 years and above, 15–59 years, and below 15 years, in that order. Hospital admissions due to demyelinating diseases of the central nervous system were more prevalent among the age group: 15–59 years, 60–74 years, 75 years and above, and below 15 years, in that order. Moreover, hospital admissions due to inflammatory diseases of the central nervous system were more prevalent among the age group: below 15 years, 60–74 years, 75 years and above, and 15–59 years, in that order (Figure 5).

## 4. Discussion 

This is the first study to look at the burden of hospital admissions in the UK due to nervous system diseases across a 20-year period. According to a meta-analysis published in 2019, the number of deaths due to neurological disorders increased by 39% between 1990 and 2016, while DALYs due to neurological disorders increased by 15% [2]. During the study period, a total of 7,380,741 (2.2%) hospital admission episodes for nervous system diseases were recorded in England and Wales, which is an alarming figure. Stroke was the second biggest cause of death worldwide in 2020, accounting for 11% of all deaths, while Alzheimer’s disease and other types of dementia were the seventh major cause of death globally, with women accounting for 65% of all deaths [11]. 

The outcomes of the study have significant implications for healthcare services. The rising number of persons admitted to hospitals with nervous system diseases necessitates many more resources to provide the best possible care. Apart from systemic atrophies that predominantly affect the central nervous system, all other groups showed a rise in hospitalization rates, as seen in Figure 1. The increase in admission rates, which was doubled or quadrupled in several categories, could not be supported by the population growth of roughly 15% during the same time period [10]. 

Episodic and paroxysmal disorders, which account for 37.4% of hospital admissions, remain one of the primary causes of DALYs and the second highest cause of death worldwide, notably stroke [2,11]. However, this category is comprised mainly of epilepsy, migraine, headache, stroke, transient cerebral ischemic attacks, and related syndromes, as well as sleep disorders [12]. Transient cerebral ischemic attack was the primary diagnosis for patients in this group. Transient cerebral ischemic attack is a temporary stroke that only lasts a few minutes. It occurs when blood flow is temporarily interrupted in a specific area of the brain [13]. A transient ischemic attack (TIA) can increase a person’s risk of having a stroke by up to 15% within 90 days, which is roughly twice as much as someone who has recently had a stroke [14]. Stroke was the largest cause of neurological DALYs and the second leading cause of death in 2016, accounting for 42.2% of all neurological DALYs. Migraine was the second-leading cause of neurological DALYs, accounting for over 16% of all DALYs [2]. The findings are consistent with our data, which show the greatest hospitalization rate and a 70.2% increase from 1999 to 2019.

Several risk factors, particularly the aging population, may play a role in the rise of admissions owing to nervous system disorders. According to studies, the aging population is linked to an increase in the number of people suffering from neurological conditions such as stroke [15]. Furthermore, increased healthcare services for stroke patients led in an increase in stroke survivors, resulting in an increase in the total number of cases [16]. By 2030, the number of stroke cases is expected to rise by 55%. This increase is primarily due to an increase in cases involving the elderly [16]. 

Aging is the main risk factor for nearly all neurodegenerative disorders such as Alzheimer’s, dementia and Parkinsonism [5,15]. According to a study by Hou and colleagues, Alzheimer’s disease affects about one out of every ten people over the age of 65 years, and the rate is expected to rise as people age. The fundamental issue with neurodegenerative disorders is that treatment choices are limited, and people with such conditions develop an irreversible chronic disease that places a significant financial load on the healthcare system [5]. Epilepsy in the elderly is caused primarily by stroke and dementia [17]. Epilepsy is more common in older people than it is in children and adults. The disease’s prevalence rate in the general population ranged from 2.3 to 15.9 per 1000 in high-income countries, whereas it was 3.6–15.4 per 1000 in low-income countries [18]. 

The UK population aged 65 years and beyond is expected to rise from 18.5% in 2019 to 23.9% by 2039, similar to many other countries [10]. With such a rise in the elderly population, the socioeconomic burden of age-related health conditions will rise, necessitating effective preventive or therapeutic interventions. 

Nerve, nerve root, and plexus disorders were another category that led to a high percentage of hospital admissions, accounting for 22.1% of total hospital admissions. The trigeminal nerve, nerve compressions, and mononeuropathies of the arms and legs, and brachial and other plexus illnesses are all included in this group. The most prevalent conditions in this category are carpal tunnel syndrome of the median nerve and cubital tunnel syndrome of the ulnar nerve [19]. Carpal tunnel syndrome has received special attention because it causes work disability and has been the subject of multiple studies because it has the potential for substantial socioeconomic consequences, especially if surgical treatment is required or if it interferes with an individual’s ability to work [20,21]. Carpal tunnel syndrome is far more common than the other disorders, according to a 2015 study conducted in the United Kingdom, and it is more common in women than in men [22]. These findings are consistent with the findings of this study (Figure 4), which found that females had a greater hospitalization rate than men in this category, where carpal tunnel syndrome is the most common. The hospitalization rate increased by 49.2%, which is consistent with the annual increase in the incidence rate [22]. Carpal tunnel syndrome was most common in women and males over the age of 75 years, according to the incidence rate by age group. However, the incidence rate and hospitalization data are compared. 

Obesity and diabetes are two key risk factors associated with an increase in carpal tunnel syndrome incidence and hospitalization [23]. Furthermore, as a result of increased labor productivity and the widespread use of computers in industry, offices, and homes, new risk concerns have emerged [21,24]. 

Demyelinating illnesses of the central nervous system accounted for 9.3% of all admissions, with a 160.3% increase over the previous two decades. Multiple sclerosis is the most common disease in this category. Hospitalization is more common in this age group in women than in males, and it is most prominent in those aged 15 to 59 years. This increase in hospitalization corresponds to the prevalence rate of multiple sclerosis, which is 190 people per 100,000 in the United Kingdom. The prevalence of the disease in women is nearly double that of men (272 versus 106 per 100,000) [25]. However, the prevalence rate of multiple sclerosis differed by sex, with women in their 50s and 60s three times more likely than males to develop the disease (578 versus 184 per 100,000 individuals). Multiple sclerosis has become more prevalent in the United Kingdom, with approximately 5200 new cases identified each year. Smoking and pre-disease obesity, especially at a young age, are important risk factors for multiple sclerosis [25,26]. Various environmental factors, such as vitamin D and sun exposure, night shift employment, and smoking, have also been linked to the development of multiple sclerosis [27]. However, there are chances to raise public health knowledge about smoking and obesity management, as well as the management of multiple sclerosis, in order to lessen the burden of this condition.

The clinical, economical, and research ramifications of this study were underlined. First, as people live longer, the social and financial burden of caring for patients with neurodegenerative diseases will undoubtedly increase. Because the majority of diseases affecting the neurological system have no cure, treatment, extensive research, and increased public funding are required to lessen the growing burden of such ailments [28]. Second, there are some commonalities among neurodegenerative illnesses, particularly risk factors, notwithstanding their differences. As a result, public awareness initiatives addressing important risk factors are necessary. The WHO stressed the importance of prevention in combating the dementia epidemic [29]. A five-year delay in the onset of dementia reduces the burden of Alzheimer’s by 50% [30]. Finally, as the burden of neurological illnesses grew, it became clear that neurological services and resources were in short supply. Furthermore, data suggest that politicians and healthcare practitioners may be unprepared to deal with the expected rise in the prevalence of neurological illnesses and impairment as a result of population aging [4,31,32]. The WHO issued recommendations in this regard, recommending that long-term therapy of chronic neurological disorders be integrated into primary care [16,32]. 

To the best of our knowledge, this is the first investigation into hospital admission patterns related to nervous system diseases in England and Wales. A clear picture of the hospitalization profile was provided by our research, which provided precise hospital admission rates over a 20-year period. This study has some limitations. The nature of the data these databases provided (data on the population level) prevented us from accessing patient-level data to identify other risk factors such as the presence of comorbidities that are connected to hospitalizations for nervous system diseases. Hospital admission data for this study include both emergency and elective hospitalizations. Therefore, our findings should be interpreted carefully.

## 5. Conclusions

In the United Kingdom, hospital admissions for nervous system diseases are on the rise. Future studies should focus on determining the factors that increase the risk of complications in patients with episodic and paroxysmal disorders, disorders of the nerve roots and plexuses, and demyelinating diseases of the central nervous system. As a result, high-risk populations will be more easily identified, and practical interventions to lower the prevalence of these disorders will be recommended.

## Figures and Tables

**Figure 1 healthcare-10-01670-f001:**
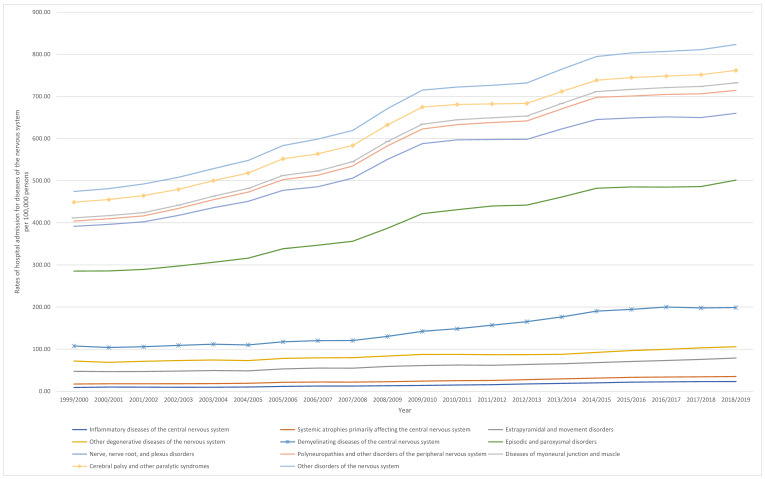
Rates of hospital admission for diseases of the nervous system in England and Wales stratified by type between 1999 and 2019.

**Figure 2 healthcare-10-01670-f002:**
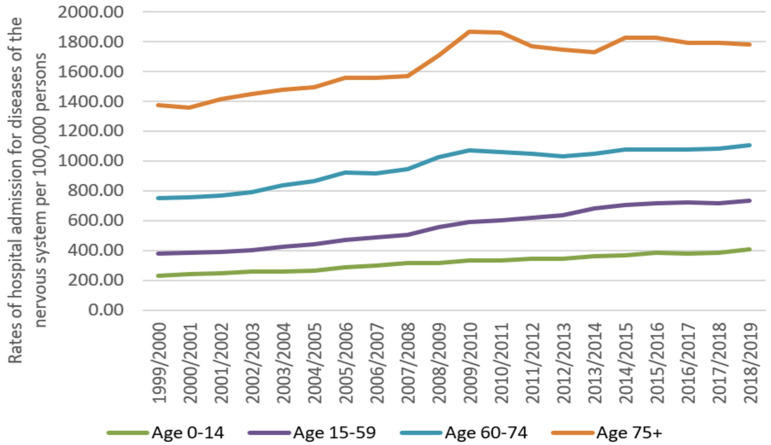
Rates of hospital admission for diseases of the nervous system in England and Wales stratified by age group.

**Figure 3 healthcare-10-01670-f003:**
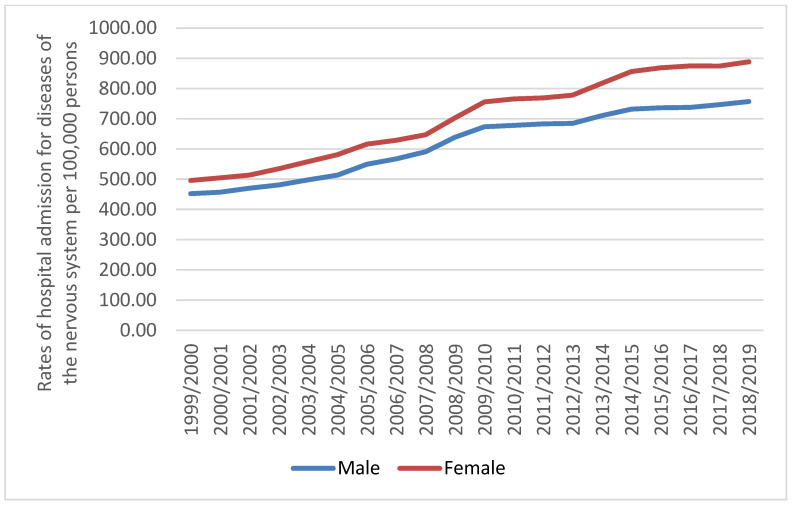
Rates of hospital admission for diseases of the nervous system in England and Wales stratified by gender.

**Figure 4 healthcare-10-01670-f004:**
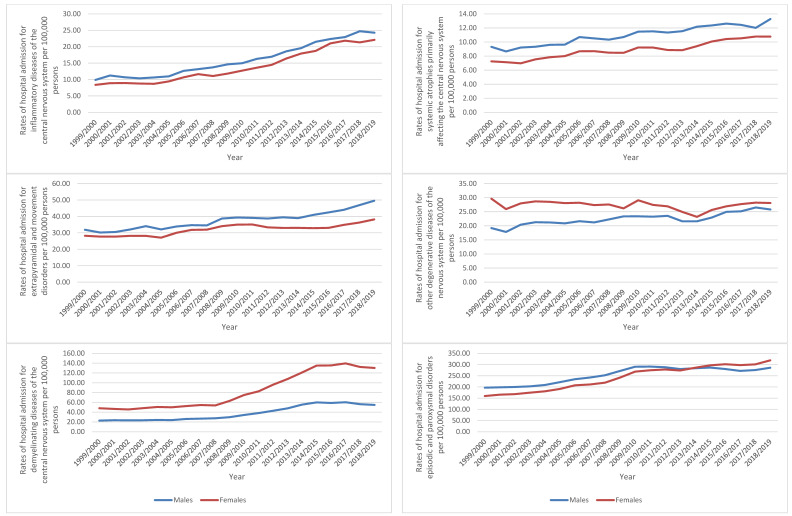
Hospital admission rates for diseases of the nervous system in England and Wales stratified by gender.

**Figure 5 healthcare-10-01670-f005:**
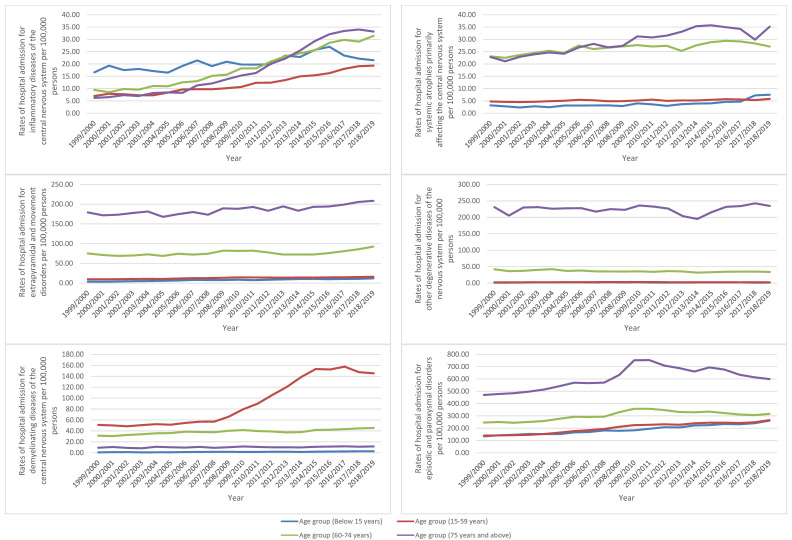
Hospital admission rates for diseases of the nervous system in England and Wales stratified by age group.

**Table 1 healthcare-10-01670-t001:** Percentage of diseases of the nervous system hospital admission from total number of admissions per ICD code.

ICD Code	Description	Number of Admissions	Percentage from Total Number of Admissions
G00–G09	Inflammatory diseases of the central nervous system	168,074	2.3%
G10–G14	Systemic atrophies primarily affecting the central nervous system	110,205	1.5%
G20–G26	Extrapyramidal and movement disorders	387,080	5.2%
G30–G32	Other degenerative diseases of the nervous system	276,408	3.7%
G35–G37	Demyelinating diseases of the central nervous system	688,479	9.3%
G40–G47	Episodic and paroxysmal disorders	2,757,086	37.4%
G50–G59	Nerve, nerve root, and plexus disorders	1,632,271	22.1%
G60–G65	Polyneuropathies and other disorders of the peripheral nervous system	385,998	5.2%
G70–G73	Diseases of myoneural junction and muscle	128,052	1.7%
G80–G83	Cerebral palsy and other paralytic syndromes	382,065	5.2%
G89–G99	Other disorders of the nervous system	465,232	6.3%

International Statistical Classification of Diseases and Related Health Problems 10th Revision (ICD-10) [3].

**Table 2 healthcare-10-01670-t002:** Percentage change in the hospital admission rates for diseases of the nervous system from 1999–2019 in England and Wales.

Diseases	Rate of Diseases in 1999 per 100,000 Persons (95% CI)	Rate of Diseases in 2019 per 100,000 Persons (95% CI)	Percentage Change from 1999–2019	*p*-Value
Inflammatory diseases of the central nervous system	9.11 (8.85–9.37)	23.17 (22.79–23.56)	154.4%	*p ≤* 0.001
Systemic atrophies primarily affecting the central nervous system	8.26 (8.01–8.51)	12.01 (11.73–12.29)	45.4%	*p ≤* 0.01
Extrapyramidal and movement disorders	30.01 (29.54–30.48)	43.80 (43.27–44.33)	46.0%	*p ≤* 0.01
Other degenerative diseases of the nervous system	24.52 (24.10–24.95)	26.95 (26.53–27.37)	9.9%	*p ≤* 0.05
Demyelinating diseases of the central nervous system	35.69 (35.18–36.21)	92.90 (92.13–93.68)	160.3%	*p ≤* 0.001
Episodic and paroxysmal disorders	177.81 (176.67–178.96)	302.59 (301.19–303.99)	70.2%	*p ≤* 0.01
Nerve, nerve root, and plexus disorders	106.25 (105.36–107.13)	158.49 (157.48–159.51)	49.2%	*p ≤* 0.01
Polyneuropathies and other disorders of the peripheral nervous system	12.50 (12.20–12.81)	54.48 (53.88–55.07)	335.6%	*p ≤* 0.001
Diseases of myoneural junction and muscle	7.62 (7.38–7.86)	17.95 (17.61–18.29)	135.5%	*p ≤* 0.001
Cerebral palsy and other paralytic syndromes	37.30 (36.78–37.83)	29.44 (29.01–29.88)	−21.1%	*p ≤* 0.05
Other disorders of the nervous system	25.37 (24.94–25.80)	61.58 (60.95–62.21)	142.7%	*p ≤* 0.001

## Data Availability

Publicly available datasets were analyzed in this study. These data can be found here: http://http//content.digital.nhs.uk/hes, http://www.infoandstats.wales.nhs.uk/page.cfm?pid=41010&orgid=869 (access on 28 June 2022).

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
