# Peer review of "An Ecological Analysis of Hospitalization Patterns for Diseases of the Nervous System in England and Wales over the Last 20 Years"

_healthcare, 2022, doi:10.3390/healthcare10091670_

Round 1

Reviewer 1 Report

The manuscript aims to provide a comprehensive overview of the hospitalization pattern of central nervous system disorders from 1999 to 2019. Such information is of great interest from a public health perspective, especially in the context of challenges for the health systems related to the aging population. However, minor revisions are needed.

Detailed comments:

Title

·        It is worth rephrasing the title in terms of hospitalizations due to Diseases of the nervous system (G00-G99) and not just central nervous system diseases

Abstract:

·        Add ICD-10 codes to each of the listed groups of nervous system diseases

Introduction:

·        Use the proper name for the ICD-10 - International Statistical Classification of Diseases and Related Health Problems 10th Revision (ICD-10) instead of ‘International Classification of Diseases system (ICD)10’ and ‘International Statistical Classification of Diseases (ICD) system’ used in this manuscript,

·        add the full name for the abbreviation – YLD

·        Remove references 8-15 - not relevant to the area of analysis undertaken and are publications of the authors of this manuscript

Methods:

·        Use the proper name for the ICD-10 - International Statistical Classification of Diseases and Related Health Problems 10th Revision (ICD-10) ) instead of ‘10th edition of the International Statistical Classification of Diseases (ICD) system’

Results:

·        Add ICD-10 codes to each of the listed groups of nervous system diseases

·        Table 1 - add numbers of hospitalization cases

·        Add p-values indicating the presence or absence of statistical significance

·        Use the proper name for the ICD-10 - International Statistical Classification of Diseases and Related Health Problems 10th Revision (ICD-10) – table 1

·        The colors in figure 1 are very difficult to distinguish

·        Please check the titles of the figures and their place in the manuscript – for egzample: Figure 2. Rates of hospital admission for diseases of the nervous system in England and Wales stratified by age group.

Discussion

·        During the study period, a total of 7,380,741 hospital admission episodes for nervous system dis-eases were recorded in England and Wales,.  What percentage of all hospitalizations is this?

Conclusions:

·        Conclusions are too short; recommendations from the study's analysis are missing

References:

·        Remove references 8-15 - not relevant to the area of analysis undertaken and are publications of the authors of this manuscript

Author Response

First of all, we would like to thank the reviewer for the time and efforts in reviewing our study and for the valuable feedback. Thank you for the opportunity to revise and resubmit our manuscript again based on the reviewers’ comment. Please find below our itemized point-by-point responses to the reviewers’ comment. Answers are written below and edited text has been highlighted (tracked changes) in the main manuscript.

Reviewer 1
The manuscript aims to provide a comprehensive overview of the hospitalization pattern of central nervous system disorders from 1999 to 2019. Such information is of great interest from a public health perspective, especially in the context of challenges for the health systems related to the aging population. However, minor revisions are needed.

 Detailed comments:

Title

  • It is worth rephrasing the title in terms of hospitalizations due to Diseases of the nervous system (G00-G99) and not just central nervous system diseases

- Thank you for this comment. We have now addressed this comment in the title in page 1.

Abstract:

  • Add ICD-10 codes to each of the listed groups of nervous system diseases

- Thank you for this comment. We have now addressed this comment in page 1.

Introduction:

  • Use the proper name for the ICD-10 - International Statistical Classification of Diseases and Related Health Problems 10th Revision (ICD-10) instead of ‘International Classification of Diseases system (ICD)10’ and ‘International Statistical Classification of Diseases (ICD) system’ used in this manuscript,

- Thank you for this comment. We have now addressed this comment throughout the manuscript.

  •  

  -  add the full name for the abbreviation – YLD

- Thank you for this comment. We have now addressed this comment in page 2, lines 55-56.

  • Remove references 8-15 - not relevant to the area of analysis undertaken and are publications of the authors of this manuscript

- Thank you for this comment. We have now addressed this comment.

Methods:

  • Use the proper name for the ICD-10 - International Statistical Classification of Diseases and Related Health Problems 10th Revision (ICD-10) ) instead of ‘10th edition of the International Statistical Classification of Diseases (ICD) system’

- Thank you for this comment. We have now addressed this comment throughout the manuscript.

Results:

  • Add ICD-10 codes to each of the listed groups of nervous system diseases

- Thank you for this comment. We have now addressed this comment in page 2, lines 84-91.

  • Table 1 - add numbers of hospitalization cases

- Thank you for this comment. We have now addressed this comment in page 3, Table 1.

  • Add p-values indicating the presence or absence of statistical significance

- Thank you for this comment. We have now addressed this comment in page 6, Table 2.

  • Use the proper name for the ICD-10 - International Statistical Classification of Diseases and Related Health Problems 10th Revision (ICD-10) – table 1

- Thank you for this comment. We have now addressed this comment in page 3, Table 1.

  • The colors in figure 1 are very difficult to distinguish

- Thank you for this comment. We have now addressed this comment and changed the style of similar coloured lines in page 5, Figure 1.

  • Please check the titles of the figures and their place in the manuscript – for egzample: Figure 2. Rates of hospital admission for diseases of the nervous system in England and Wales stratified by age group.

- Thank you for this comment. We have now addressed this comment throughout the manuscript.

Discussion

  • During the study period, a total of 7,380,741 hospital admission episodes for nervous system dis-eases were recorded in England and Wales,.  What percentage of all hospitalizations is this?

- Thank you for this comment. We have now addressed this comment and added the percentage (which is 2.2%) out of the total number of admissions, in page 12, line 213.

Conclusions:

  • Conclusions are too short; recommendations from the study's analysis are missing

- Thank you for this comment. We have now addressed this comment in page 14, lines 324-328.

References:

  • Remove references 8-15 - not relevant to the area of analysis undertaken and are publications of the authors of this manuscript

- Thank you for this comment. We have now addressed this comment.

Reviewer 2 Report

I have read the manuscript entitled "An ecological analysis of hospitalization patterns for central nervous system diseases in England and Wales over the last 20 years", which has been submitted for possible publication in Healthcare. The paper is generally well written and considers a relevant topic, but it also exhibits some deficiencies. An adequately revised version, where all my comments are carefully addressed, should have the potential for publication.

1    Abstract, "hospitalization pattern of such disorders": The Abstract should be self-contained and should, thus, not refer to the title by writing "such disorders".

2    6th line of Introduction: space missing in "(ICD)10".

3    Section 2.1: When referring to the ICD-10 system, reference 20 should be added, and maybe also in the caption of Table 1.

4    Section 2.2 is too vague. You state that "A Poisson model was used to examine the trend in hospital admissions.", could please provide more details?

Furthermore, you state that "Chi-square test was performed to determine the difference in hospital admission rates between 1999 and 2019."? There are many types of Chi-square tests in the literature, which one exactly did you use in which way? Are the conditions for using the Chi-square approximation satisfied?

5    The caption of Figure 2 is misplaced.

6    Section 4, third paragraph, episodic and paroxysmal disorders: In addition to your explanations, one can see a somewhat "sudden jump" in the age group 75+ between 2007 and 2010, e.g., in Figure 2. Is this jump caused by strokes? And why do we have such a sudden jump?

Author Response

First of all, we would like to thank the reviewer for the time and efforts in reviewing our study and for the valuable feedback. Thank you for the opportunity to revise and resubmit our manuscript again based on the reviewers’ comment. Please find below our itemized point-by-point responses to the reviewers’ comment. Answers are written below and edited text has been highlighted (tracked changes) in the main manuscript.

I have read the manuscript entitled "An ecological analysis of hospitalization patterns for central nervous system diseases in England and Wales over the last 20 years", which has been submitted for possible publication in Healthcare. The paper is generally well written and considers a relevant topic, but it also exhibits some deficiencies. An adequately revised version, where all my comments are carefully addressed, should have the potential for publication.

1    Abstract, "hospitalization pattern of such disorders": The Abstract should be self-contained and should, thus, not refer to the title by writing "such disorders".

- Thank you for this comment. We have now addressed this comment in page 1, line 18.

2    6th line of Introduction: space missing in "(ICD)10".

- Thank you for this comment. We have now addressed this comment in page 1.

3    Section 2.1: When referring to the ICD-10 system, reference 20 should be added, and maybe also in the caption of Table 1.

- Thank you for this comment. We have now addressed this comment and added the relevant reference to section 2.1 and Table 1.

4    Section 2.2 is too vague. You state that "A Poisson model was used to examine the trend in hospital admissions.", could please provide more details?

- Thank you for this comment. We have now addressed this comment and added further details in page 3, lines 105-109.

Furthermore, you state that "Chi-square test was performed to determine the difference in hospital admission rates between 1999 and 2019."? There are many types of Chi-square tests in the literature, which one exactly did you use in which way? Are the conditions for using the Chi-square approximation satisfied?

- Thank you for this comment. We have now clarified the type of test used in page 3, line 103. Yes, the conditions of applying the test were satisfied.

5    The caption of Figure 2 is misplaced.

- Thank you for this comment. We have now addressed this comment and added the caption for it.

6    Section 4, third paragraph, episodic and paroxysmal disorders: In addition to your explanations, one can see a somewhat "sudden jump" in the age group 75+ between 2007 and 2010, e.g., in Figure 2. Is this jump caused by strokes? And why do we have such a sudden jump?

- - Thank you for this comment. The main driver for this jump was the increase in the number of admissions for transient cerebral ischaemic attack. We have now clarified this further in the discussion in page 12, lines 229-234. However, due to the nature of the study which is ecological study on the population level, we did not have data on the individual level including comorbidities, which restricted our ability to interpret our findings further. This limitation is now highlighted in the discussion section, in page 14, lines 313-321.

Round 2

Reviewer 2 Report

I have read the revised manuscript entitled "An ecological analysis of hospitalization patterns for central nervous system diseases in England and Wales over the last 20 years", which has been submitted for possible publication in Healthcare. The paper has been generally improved during revision, but there is still one issue left that has not been solved satisfactorily. So again a careful revision is necessary.

In my previous report, I asked: "There are many types of Chi-square tests in the literature, which one exactly did you use in which way? Are the conditions for using the Chi-square approximation satisfied?"

As a reaction to this comment, you simply added the word "Pearson" to the text and wrote in your reply to me that "Yes, the conditions of applying the test were satisfied.", which is (by far) not sufficient.

First, the add-on "Pearson" still does not make the test uniquely identifiable, there are various Pearson chi-square tests in the literature, e.g., regarding goodness-of-fit, independence, homegeneity, etc. So which one EXACTLY did you you? Please write into the manuscript!

Second, once you have made clear which test you used, you should write INTO THE MANUSCRIPT
(1) which conditions need to be satisfied such that the Chi-square approximation works sufficiently well,
(2) and why they are satisfied in your case!

Author Response

Manuscript ID healthcare-1854620, titled “An ecological analysis of hospitalization patterns for central nervous system diseases in England and Wales over the last 20 years”

Corresponding authors: Dr. Fatemah M. Alsaleh

Dear Editor,

Thank you for the opportunity to revise and resubmit our manuscript again based on the reviewers’ comment. Please find below our itemized point-by-point responses to the reviewers’ comment. Answers are written below and edited text has been highlighted (tracked changes) in the main manuscript.

Reviewer Comments:

Reviewer 2:

I have read the revised manuscript entitled "An ecological analysis of hospitalization patterns for central nervous system diseases in England and Wales over the last 20 years", which has been submitted for possible publication in Healthcare. The paper has been generally improved during revision, but there is still one issue left that has not been solved satisfactorily. So again a careful revision is necessary.

In my previous report, I asked: "There are many types of Chi-square tests in the literature, which one exactly did you use in which way? Are the conditions for using the Chi-square approximation satisfied?" As a reaction to this comment, you simply added the word "Pearson" to the text and wrote in your reply to me that "Yes, the conditions of applying the test were satisfied.", which is (by far) not sufficient.

First, the add-on "Pearson" still does not make the test uniquely identifiable, there are various Pearson chi-square tests in the literature, e.g., regarding goodness-of-fit, independence, homegeneity, etc. So which one EXACTLY did you you? Please write into the manuscript!

- Thank you for this comment. We are sorry as we thought you wanted us to write it in general. We used Pearson Chi-square test for independence. We have now added this to the statistical analysis part in the manuscript in page 3, line 100.

Second, once you have made clear which test you used, you should write INTO THE MANUSCRIPT

(1) which conditions need to be satisfied such that the Chi-square approximation works sufficiently well,

(2) and why they are satisfied in your case!

- Thank you for this comment. We have now clarified in the manuscript why we have chosen Pearson Chi-square for independence and how it is satisfied in our study in the manuscript in page 3, lines 100-102.